# Chitosan-Polyaniline (Bio)Polymer Hybrids by Two Pathways: A Tale of Two Biocomposites

**DOI:** 10.3390/polym16182663

**Published:** 2024-09-22

**Authors:** Yuriy A. Anisimov, Heng Yang, Johnny Kwon, Duncan E. Cree, Lee D. Wilson

**Affiliations:** 1Department of Chemical Engineering, McMaster University, Hamilton, ON L8S 4M6, Canada; anisimoi@mcmaster.ca; 2Department of Chemistry, University of Saskatchewan, 110 Science Place, Saskatoon, SK S7N 5C9, Canada; 3Department of Mechanical Engineering, McMaster University, Hamilton, ON L8S 4L8, Canada

**Keywords:** polyaniline-chitosan composites, covalent bonding, noncovalent interactions, structure-function relationships

## Abstract

Previous research highlights the potential of polyaniline-based biocomposites as unique adsorbents for humidity sensors. This study examines several preparative routes for creating polyaniline (PANI) and chitosan (CHT) composites: Type 1—in situ polymerization of aniline with CHT; Type 2—molecular association in acidic aqueous media; and a control, Type 3—physical mixing of PANI and CHT powders (without solvent). The study aims to differentiate the bonding nature (covalent vs. noncovalent) within these composites, which posits that noncovalent composites should exhibit similar physicochemical properties regardless of the preparative route. The results indicate that Type 1 composites display features consistent with covalent and hydrogen bonding, which result in reduced water swelling versus Type 2 and 3 composites. These findings align with spectral and thermogravimetric data, suggesting more compact structure for Type 1 materials. Dye adsorption studies corroborate the unique properties for Type 1 composites, and ^1^H NMR results confirm the role of covalent bonding for the in situ polymerized samples. The structural stability adopts the following trend: Type 1 (covalent and noncovalent) > Type 2 (possible trace covalent and mainly noncovalent) > Type 3 (noncovalent). Types 2 and 3 are anticipated to differ based on solvent-driven complex formation. This study provides greater understanding of structure-function relationships in PANI-biopolymer composites and highlights the role of CHT as a template that involves variable (non)covalent contributions with PANI, according to the mode of preparation. The formation of composites with tailored bonding modalities will contribute to the design of improved adsorbent materials for environmental remediation to versatile humidity sensor systems.

## 1. Introduction

The synthesis of polyaniline (PANI)/chitosan (CHT) composites dates back several decades, where “chitaline” (CHT + PANI) was first reported by Yang et al. [1]. They reported that covalent grafting occurs between PANI and CHT, which was concluded on the basis of limited spectral evidence. Recently [2], “chitaline” was cited several times, where the key structural evidence for covalent bonding in such PANI/CHT composites was based on IR spectroscopy. The synthesis and properties of PANI/CHT composites were reported earlier, where unique features such as catalyst supports with high efficiency led to the catalytic reduction process of 4-nitrophenol [3], along with unique adsorption properties toward cationic dyes such as methylene blue [4].

To date, CHT is inferred to serve a dual role in the formation of PANI composites, acting both as a noncovalent template [5] and a reactive substrate [3]. As a passive template, CHT directs the morphology of PANI, where the uniform distribution of PANI particles across the CHT matrix is observed [5]. This observation implies that the presence of CHT favours the formation of highly ordered structures due to its role as a supramolecular template, which contrasts with the irregular nanosized particles without a template [6].

Furthermore, the role of CHT as a reactive substrate facilitates copolymer formation, resulting in increased entanglement and enhanced complexation between CHT and PANI. Previous work highlights strong electronic PANI-CHT interactions that vary with the weight ratios of CHT and aniline during in situ polymerization [3]. This suggests that the bonding between CHT and PANI, which may include covalent interactions at specific stoichiometry’s, plays a crucial role in defining the composite properties. This interaction modifies the microenvironment during polymerization, leading to enhanced crosslinking, and complementary stabilization within the composite structure.

The role of CHT as a component in nanocomposites extends beyond its structural and reactive function, that significantly influence its mechanical properties. A recent study has shown that varying the atom ratio of CHT can markedly alter the mechanical performance of silica aerogel/CHT nanocomposites, enhancing mechanical properties (e.g., ultimate strength and Young’s modulus) [7]. These findings suggest that similar effects could potentially be translated to other composite systems, including those involving PANI, where CHT may equally contribute to enhanced structural integrity and mechanical performance.

In order to gain insight on the structure-function relationships for such systems, there is a need to carry out systematic structural studies on PANI/CHT composites. The growing interest in these PANI-based systems is marked by continued research on its electrical conductivity [8], analogous to conductive polymers like polyacetylene. Thus, the need for alternative materials with improved properties [9], such as PANI, relate to its variable oxidation states, acid-doped and undoped forms, along with its diverse morphology induced by structural templates [10].

Despite PANI’s remarkable conductivity [11], a complete understanding of its moisture adsorption properties is somewhat limited [12], especially for the case of composite materials. While CHT is a non-conductive biopolymer with abundant polar groups (–OH and –NHR; where R = H or acetyl), there are studies that indicate CHT functions as a structural template during polymerization of aniline (ANI). In turn, the presence of CHT appears to alter the hydrophile-lipophile balance (HLB) and the mechanical strength of PANI [13]. Intuition suggests that the use of non-conductive biopolymers together with PANI would lower its conductivity. However, the opposite effect occurs upon the formation of binary composites (CHT + PANI) under oxidative conditions. Polymerization of aniline (ANI) in the presence of CHT yields unique conductivity and other properties that are suitable for humidity sensing [13]. Protonation-deprotonation and redox switching of PANI provide the chemical basis of humidity sensing [14]. Composite formation between PANI and CHT is inferred to result in covalent and/or hydrogen bonding (HB) between the donor-acceptor (D-A) groups (OH/NH/NH_2_) between the composite components. Covalent bonding may occur upon condensation between reactive (OH/NH_2_) groups. In turn, PANI/CHT composites ought to promote a proton-hopping mechanism where H+ can migrate along and across (bio)polymer chains. In this context, H_2_O serves as the proton-carrier (Figure 1a,b), which affects the overall observed conductivity.

The above mechanism in Figure 1 provides motivation to characterize the structure and interactions (noncovalent vs. covalent) between the D-A groups of PANI and CHT in the present study. The novelty of the current research lies in the study and comparison of three distinct preparative routes for PANI/CHT composites. Unlike previous studies, which typically focus on a single synthesis method, this work systematically investigates how these different routes influence the formation and extent of bonding between PANI and CHT. This aspect could not be fully understood through a single preparative route. Acquiring structural insight about the PANI/CHT composites can highlight the role of HB with water and proton transfer for such materials. The PANI-CHT hydrogen bonds can be disrupted by water, where weaker interactions occur between the polymer backbones versus the pair-wise HB interactions of H_2_O [15]. In Figure 1, the uptake of water will dissociate the PANI/CHT noncovalent complex, accompanied by swelling. By contrast, the presence of covalent bonds, along with favourable electrostatic interactions, affect the swelling of PANI/CHT composites upon water uptake. [2] The proton transfer (Figure 1) should occur to a greater extent in noncovalent composites, leading to greater conductivity, especially for hydrated materials. We posit that Type 1 and 2 composites will yield binary composites with structural variability due to the presence of favourable D–A interactions, influenced by the type of bonding between CHT and ANI/PANI. The goal of the current research relates to the preparation of PANI/CHT composites via several preparative routes: (1) in situ polymerization, (2) molecular association and (3) physical blending. Route 3 provides reference to a powder physical mixture (without liquid-assisted grinding) that is unlikely to introduce any covalent bonding. In light of these goals, it is posited that a comparison of the binary composites (Types 1–3) will provide insight on the structure–function relationships of PANI/CHT composites (*cf.* Figure 1). To address the hypothesis, several objectives are addressed: (1) To prepare PANI/CHT composites (Types 1–3) by various synthetic routes; (2) To carry out structural and physicochemical characterization of the binary composites using spectroscopic and complementary methods. In turn, this study will address knowledge gaps related to the nature of the bonding (physical/noncovalent vs. covalent) between the PANI/CHT components and to characterize the structure of the various binary biocomposites (Types 1–3). The structure-function relationships of the (bio)polymers for such systems will contribute to the design of advanced materials with improved properties for diverse applications, such as adsorbents for environmental contaminants, humidity sensor materials, and redox active catalysts.

## 2. Materials and Methods

### 2.1. Materials

Aniline (ACS reagent grade, 99.5% purity) with a density of 1.02 g/cm^3^ and CHT powder (degree of deacetylation 80%, low molecular weight), as well as ammonium persulfate (APS) with 98% purity (ACS reagent grade), were obtained from Sigma-Aldrich, Oakville, ON, Canada. Hydrochloric acid (HCl) 36% *w*/*w* aqueous solution was purchased from Alfa Aesar, Tewksbury, MA, USA. Methylene blue (MB) of the highest available purity was procured from Sigma-Aldrich, Oakville, ON, Canada. Milli-Q water of high purity and low resistivity (17–18 MΩ·cm) was used for preparation of all solutions. Potassium bromide (KBr, high purity spectral grade) was obtained from Sigma-Aldrich (Oakville, ON, Canada) and was used for recording the FTIR spectra.

### 2.2. Shorthand Notations

The composition of all samples was referenced to the PANI content (%). For example, 25% PANI refers to a binary composite containing 25 wt.% PANI and 75 wt.% CHT. As well, each composite was demarcated according to its processing route (i.e., in situ polymerization (Route 1), molecularly associated (Route 2), and physical blending (Route 3). For example, 75% PANI in situ refers to a sample containing 75% PANI, 25% CHT that was prepared by the in situ polymerization method.

### 2.3. Preparation of Polymer Composites via Three Different Routes

Three composites were prepared with different ratios between PANI and CHT (25/75, 50/50 and 75/25), as illustrated in Figure 2.

The first set, Type 1 was prepared via in situ polymerization (Route 1), where the aniline monomer was added to acidic CHT solution. Type 2 composites were prepared via a molecular association method using acidic media (Route 2) to enable association of PANI and CHT upon mixing, while Type 3 composites involved physical blending of PANI and CHT powders without solvent (Route 3).

### 2.4. Synthesis of Composites

#### 2.4.1. In Situ Polymerized Composites—Route 1

An accurately measured quantity of CHT (from 1 to 3 g depending on the ratio to aniline) was dissolved in 1 M HCl (100 mL) and placed in an ultrasonic bath for 1 h. The system was then cooled to 0 °C with an ice bath, and aniline was added dropwise with a syringe. Subsequently, an acidic solution of ammonium persulfate (APS) (100 mL, 1 M HCl) was introduced gradually using a burette over a 30 min period, maintaining an APS molar ratio of 0.8:1. The solution colour transitioned to dark blue during the addition. After the APS was fully added, the mixture was stirred for an additional hour, followed by neutralization with NaOH. The resultant precipitate was collected by filtration, thoroughly washed with deionized water, and dried in an oven at 60 °C overnight. To remove any remaining APS and NaOH, the composite was subjected to an additional washing step and then dried for another 24 h [13].

#### 2.4.2. Molecularly Associated Composites—Route 2

In two separate beakers, appropriate amounts of CHT and PANI were dissolved in HCl and left in an ultrasonic bath for 1–2 h. Then, the mixture was continuously stirred for 24 h. Thereafter, the solution was brought to pH 8 with NaOH until a deep purple residue formed upon precipitation. The product was filtered and collected in a Petri dish, with subsequent heating at 50 °C until the product was dry.

#### 2.4.3. Physically Blended Samples—Route 3

In two separate mortars, appropriate amounts of CHT and PANI were, respectively, ground to achieve a 75 µm mesh size. The powders were then combined and thoroughly mixed to produce a uniform blend, where the products were stored in vials for further use.

### 2.5. Materials Characterization

#### 2.5.1. ^13^C Solid-State NMR Spectroscopy

^13^C solid-state NMR spectra were recorded using a Bruker AVANCE III HD 500 MHz NMR spectrometer operating at 125.76 MHz frequency. Samples were packed into 4 mm drop-in Si_3_N_4_ rotors. Spectral analysis was conducted with a pulse sequence that employed Cross Polarization along with a Total Suppression of Spinning Sidebands (CP-TOSS) at a spinning frequency of 7.5 kHz with a 1 ms cross-polarization time. Spectral acquisition employed a variable number of scans (4096 to 8192) with a recycle delay of 2 s. Adamantane (δ = 38.48 ppm) was the external reference for ^13^C chemical shifts (δ). The NMR spectral results were processed using Topspin 4.0.7 Bruker NMR software.

#### 2.5.2. Thermogravimetric Analysis (TGA)

Thermal analysis of the materials was conducted using a TA Instruments Q50 TGA system (TA instruments, New Castle, DE, USA). The heating rate was set to 5 °C per minute from 25 °C to 495 °C with N_2_ carrier gas. The thermogram profiles portray the first derivative (DTG) plots (weight change (%/°C) vs. temperature (°C)) to analyze the thermal stability of the composites.

#### 2.5.3. Scanning Electron Microscopy (SEM)

The sample surface morphology was surveyed using a Jeol JSM-6010LV scanning electron microscope (Tokyo, Japan) at an accelerated voltage of 10 kV. The SEM instrument was operated at a high vacuum imaging mode using the secondary electron image (SEI) detector. The biopolymer composites were gold coated prior to SEM imaging using an Edwards S150B sputter coater (Crawley, West Sussex, UK). A thin layer of gold imparts conductivity to the organic samples since they were initially non-conductive.

#### 2.5.4. Solvent Swelling Tests

To ensure consistency in the swelling tests, all samples were sieved using a 75 µm mesh prior to analysis. This procedure standardizes the sample particle size, which minimizes any potential variable swelling behaviour due to differences in particle size distribution.

Swelling tests in pure water were carried out in accordance with the procedure described elsewhere [13]. The degree of swelling (*S_W_*; %) was calculated according to Equation (1):(1)$$S_{w}=\frac{m_{w}-m_{d}}{m_{d}}\times 100\%$$
where $m_{w}$ is the weight of the swollen sample, and $m_{d}$ is the dry weight.

Standard deviations were calculated according to Equation (2) based on measurements replicated five times:(2)$$\sigma =\sqrt{\frac{{\sum_{i=1}^{n}(S_{wi}-\bar{S_{w}})}^{2}}{N-1}}$$
where σ is the standard deviation, $S_{wi}$ is swelling (%) of *i* sample, $\bar{S_{w}}$ is a mean swelling value. The absolute swelling errors ($∆S_{w}$) were calculated according to Equation (3):(3)$$∆S_{w}=\frac{3\sigma }{\sqrt{N}}$$
where N is number of measurements for Equations (2) and (3).

#### 2.5.5. Fourier Transform Infrared (FTIR) Spectroscopy

IR spectra of powdered samples were obtained using a Bio-RAD FTS-40 spectrophotometer (Bio-RAD Laboratories, Inc., Hercules, CA, USA). Samples were blended with KBr (FTIR grade) in a 1:10 wt. ratio (*w*/*w*; sample: KBr) with subsequent grinding in a mortar and pestle. The diffuse reflectance infrared Fourier transform (DRIFT) spectra were recorded in reflectance mode from 4000 to 700 cm^−1^ with multiple scans (*n* = 256) and spectral resolution of 4 cm^−1^. A baseline correction was performed using KBr as a reference. All spectra were processed and normalized using Renishaw Wire 5.2 software.

#### 2.5.6. Raman Spectroscopy

Raman spectra of solid samples were recorded with a Renishaw inVia reflex Raman microscope (Renishaw (Canada) Limited, Mississauga, ON, Canada) using a 785 nm solid-state diode laser (500 mW). The grating system was set to 1200 lines/mm at 5% laser power. The sample was analyzed using a microscope equipped with a Leica objective with 20× magnification (numerical aperture = 0.40). The backscattered Raman signal collection was acquired with the aid of a Pelletier cooled CCD detector. The spectra were recorded from 300 to 1800 cm^−1^ with a 2 cm^−1^ spectral resolution. A fixed number of scans (*n* = 16) and the instrument calibration employed an internal Si (110) reference at 521 cm^−1^. All spectra were baseline-corrected and normalized using the Renishaw Wire 5.2 software.

#### 2.5.7. Powder X-ray Diffraction (PXRD) Analysis

XRD data collection employed a Rigaku Ultima IV X-ray diffractometer (Rigaku Americas, The Woodlands, TX, USA) with monochromatic Cu Kα radiation source (λ = 0.154 nm). The applied voltage and current was set at 45 kV and 40 mA, respectively. The powders were placed in a sample holder, where they formed a smooth horizontal sample surface for the incident and reflected beam. PXRD spectra were measured continuously over a 2θ range (5−70°) with a scan rate of 0.5°/min.

#### 2.5.8. Equilibrium Dye Adsorption

Adsorption profiles were obtained in batch mode (*n* ≈ 15) for powdered samples with a fixed mass (20.0 ± 0.4 mg). The samples were placed in vials which were filled with methylene blue (MB) dye solution. The MB dye concentration ranged from 5 to 400 µM. Each set of solutions were placed onto a horizontal shaker at 125 rpm for 24 h. The dye concentration before and after adsorption was estimated according to Beer’s law based on the optical absorbance of MB solutions using a Varian Cary 100 Scan UV-Vis spectrophotometer (Agilent Technologies, Inc., Santa Clara, CA, USA). Calibration curves for MB employed a fixed λ_max_ (664 nm). Equilibrium dye uptake $\left(q_{e};\;µmol/g\right)$, was calculated according to Equation (4) [16]:(4)$$q_{e}=\frac{m(C_{0}-C_{e})}{V}\;$$
where m is the weight (mg) of adsorbent, *V* is volume of MB solution (mL), $C_{0}$ and $C_{e}$ are the initial and final MB concentrations (µM), respectively.

The adsorption isotherms were analyzed according to several models, as described elsewhere [13]. The Sips isotherm model [17] affords calculation of dye uptake at equilibrium, according to Equation (5):(5)$$q_{e}=\frac{q_{m}{K_{s}C_{e}}^{n_{s}}}{1+{K_{s}C_{e}}^{n_{s}}}$$
where $q_{m}$ is the monolayer adsorption capacity of the dye at saturated conditions, $K_{s}$ is the Sips isotherm model constant, $C_{e}$ is the equilibrium dye concentration in solution, and $n_{s}$ is the Sips isotherm exponent (heterogeneity factor) term.

The Sips isotherm is among the most common models to describe solid–liquid adsorption processes to provide insight on the adsorption mechanism. A versatile theory was developed by Dubinin and Radushkevich (D-R) [18], where surface heterogeneities can be classified and attributed to a certain type of adsorbent. Although it was initially designed for adsorption of organic vapours onto activated carbons, it is also effective for liquid-phase adsorption [19,20,21,22,23]. The D-R model (Equation (6)) provides an estimation of the adsorption capacity of the adsorbent:(6)$$q_{e}=q_{m}e^{-{\left(\frac{\varepsilon }{E}\right)}^{2}}$$
where $q_{e}$ and $q_{m}$ are defined as described by Equation (5), *ε* is the adsorption potential (J/mol), and *E* is the effective adsorption energy (binding energy, J/mol), which is a constant that characterizes the adsorbent–adsorbate interactions. The adsorption potential *ε* is thermodynamically equal to the negative free energy of the adsorption process that can be calculated according to Equation (7):(7)$$\varepsilon =RTln\frac{C}{C_{e}}$$
where R is the gas constant (8.314 J/mol·K), T is temperature, C and $C_{e}\;$are the maximum and equilibrium concentrations of dye solution in SI units, respectively.

The Dubinin-Radushkevich (D-R) model is a useful tool for the calculation of the binding energy of an adsorbate to an adsorbent (E), according to Equation (6). However, there is a limitation since this model is valid only for homogeneous adsorbents containing the same pore geometry and size. Later, it was modified into the Dubinin–Astakhov (D-A) model [24], according to Equation (8):(8)$$q_{e}=q_{m}e^{-{\left(\frac{\varepsilon }{E}\right)}^{n}}$$
where the exponent (*n*) is a heterogeneity factor. The role of the heterogeneity factor and its relative magnitudes are described further in Section 3.2.5.

#### 2.5.9. Solubility Tests

Type 1 biocomposites and the starting materials were subjected to a solubility test in dimethyl sulfoxide (DMSO). Each sample (ca. 550 mg) was added to DMSO with a final volume (250 mL). The samples were placed onto a horizontal shaker table and allowed to equilibrate for 24 h. After that, the soluble fraction was separated from the insoluble residue, where the latter was vacuum-pumped at 90 °C for 72 h. Then, the residual amount of soluble fraction was calculated as a sample solubility in mg/g.

#### 2.5.10. Proton Nuclear Magnetic Resonance (^1^H NMR) Spectroscopy

^1^H NMR spectra were recorded at 500 MHz frequency using a wide-bore (89 mm) probe equipped with a 5 mm PATX1 probe and a 11.7 T Oxford superconducting magnet. A SSSC 500 console and workstation running X-WIN NMR 3.5 (Bruker Bio Spin Corp.; Billerica, MA, USA) were used to control the operating parameters. All NMR spectra were processed using the TopSpin 4.0.7 software.

## 3. Results and Discussion

To gain insight on the bonding (covalent and/or noncovalent) among the structural units of PANI and CHT, complementary spectral techniques (^1^H NMR, ^13^C solids NMR, FTIR and Raman) were used to characterize the structural features of the composites, according to the synthetic routes (Figure 2). Complementary methods for characterizing the structure and physicochemical properties of the (bio)polymers (CHT, PANI) and the composites included thermogravimetry, PXRD, solvent swelling and dye adsorption in water, as described below.

### 3.1. Chemical Characterization

#### 3.1.1. Proton Nuclear Magnetic Resonance (^1^H NMR) Spectroscopy

^1^H NMR spectroscopy was used to assess the formation of chemical bonds for Type 1 composites. Figure 3 displays spectral data for both pristine materials and the composites. It is evident that the spectral regions of PANI and CHT do not overlap (5.5–8 ppm for PANI and 1–5 ppm for CHT), where the characteristic ^1^H NMR spectral signatures are listed in Table 1, which are supported by other independent studies [25,26,27]. Table 1 reports the most characteristic chemical shifts from Figure 3.

Spectral signatures that appear between 5.7 and 6.2 ppm are related to the side products of aniline oxidation, such as 1,4-benzoquinones [4,28]. The solvent (DMSO) signal is observed at 2.5 ppm for all spectra, whereas the signature for H_2_O varies from 4.8 to 5.0 ppm (pristine materials) to 5.2 ppm (for composites). This trend supports that stronger intermolecular hydrogen bonding occurs between composites and H_2_O as compared to pristine materials. The trend in chemical shifts concurs with the mechanism described in a recent review [2], where H_2_O molecules tend to be adsorbed by PANI/CHT chains of the composite. In turn, hydration results in the so-called “unzipping” process (*cf.* Figure 1b above). PANI/CHT composites display new signatures that are not observed for the starting materials (highlighted in yellow in Figure 3): δ = 7.93 ppm (–HC6=N–) and δ = 4.05 ppm (–C5H–), doublet and multiplet, respectively. The C6H proton couples with one C5H proton (^3^J_H-H_ spin-spin coupling), resulting in a doublet for C6H. The C5H proton, in turn, couples with two neighbouring protons (^3^J_H-H_ spin-spin coupling with C4H and C6H), therefore causing a doublet of doublets (*cf*. multiplet in Figure 3). This supports that covalent bonding occurs at C6 of CHT; a high chemical shift (low field) at 7.93 ppm supports the formation of a Schiff-base adduct. The reactivity of C6- over C3-OH (primary vs. secondary C-OH) is well known due to differences in acidity (pK_a_). The formation of such CHT Schiff-base species has precedence, according to other reports [29,30]. The ^13^C spectral data of the Schiff base signature for –CH=N– are expected at ~157 ppm, which is obscured by spectral overlap with the quinoid (Q) regions of PANI [31,32]. Therefore, the formation of a C6 Schiff base species was not readily detected by routine solids NMR spectral data. By comparison, evidence is revealed by ^1^H NMR spectral data in solution (*cf.* Figure 3).

It is noteworthy that the C2–C6 CHT signals for the binary composites are significantly shifted versus pristine CHT (2.9 to 3.9 ppm vs. 3.2 to 4.0 ppm). This may indicate that C2 undergoes chemical and/or physical change. The latter was not discussed above since the corresponding ^1^H signatures likely overlap with other CHT signatures. However, recent reports on PANI/CHT composites describe C2 cross-linking [33,34,35]. Thus, potential structures are illustrated in Figure 3b,c. In parallel, ring-opening reactions of CHT due to the oxidative conditions may also occur, where similar trends are reported for periodate as the oxidizing agent [36,37]. Since IO_4_^−^ may react similar as APS, the role of oxidative ring opening cannot be excluded herein.

The spectral results show that the integration of ^1^H nuclei corresponding to covalent grafts (Table 1) are much smaller than that of PANI and CHT protons, suggesting that Type 1 composites are cross-linked at relatively low levels, where additional contributions due to HB interactions impart enhanced structural stability. These findings correlate with results from other spectral techniques obtained herein (*cf.* Section 3.1.2, Section 3.1.3, Section 3.1.4, Section 3.2.1, Section 3.2.2, Section 3.2.3, Section 3.2.4, Section 3.2.5 and Section 3.2.6).

#### 3.1.2. ^13^C Solid State NMR Spectroscopy

To characterize the carbon structure of the precursors and the binary composites, ^13^C solids NMR spectra were obtained (Figure 4).

PANI in its emeraldine base (EB) form exhibits ^13^C spectral lines at 110, 123, 138, 141, 145 and 157 ppm, which concur with an earlier study [38]. CHT spectral lines appear at 23, 58, 61, 75, 83, 105 and 173 ppm agree with a recent study [39]. According to Figure 4, the binary composites display distinctive ^13^C signatures for PANI and CHT, where the spectral regions do not show any extensive overlap between the polysaccharide and benzenoid/quinoid (B/Q) units of PANI. NMR line intensities for PANI increase (and decrease proportionally for CHT) as the weight fraction of PANI increases.

There is a notable change in the chemical shift in C6 for CHT with 25% PANI for the in situ composite, as compared with pristine CHT (62.8 ppm vs. 60.5 ppm). This trend provides support for hydrogen bond formation between the –CH_2_OH groups of CHT and –NH– groups of PANI, which concur with FTIR and Raman spectral results (*cf.* Section 3.1.3 and Section 3.1.4). H-bonding preferably occurs for C6-OH since it is more acidic than C3-OH. Interestingly, this change was not observed for 50% and 75% PANI for the in situ composites, which suggests a unique composite structure containing a lower (25%) weight fraction of PANI, which can be attributed to greater enclathration of PANI by CHT. In contrast, molecularly associated composites and physical blends did not show this change. Another remarkable feature of the 25% PANI in situ polymerized material was the presence of two signatures for the –CH_3_ groups (23.6 and 30.3 ppm), which may also relate to covalent cross-linking effects between polymer subunits or inductive effects of the amide carbonyl upon HB interactions with PANI. Alternatively, the –CH_3_ group may undergo oxidization due to APS, which can result in downfield chemical shifts.

Additionally, Figure 4c reveals a small signal at 30.3 ppm for Route 2, with reduced intensity compared to Route 1. This small signature may suggest minor residual covalent bonding or inductive effects, as described above. However, such interactions are present only in negligible amounts, which concur with the role of inductive effects via H-bonding between PANI and CHT. The predominant interactions in Route 2 are primarily non-covalent, as corroborated by the IR spectral data (*cf.* Section 3.1.3 and Section 3.1.4).

The shoulder of the ^13^C6 NMR line vanishes and becomes less pronounced at higher PANI content (%). This trend implies that potential hydrogen bonding between CHT and PANI via C6-OH becomes dilute as the PANI fraction increases. This effect arises due to changes in cross-polarization efficiency because of differences in the motional dynamics of HB vs. non-HB systems. The minor peaks at 30.3 ppm for 25% PANI (aliphatic carbon); 96.2 ppm (C–O fragment) and 180.2 ppm (-C=O) for 75% PANI are not visible for the starting materials. This suggests that 25% and 75% PANI samples undergo specific chemical interactions that are not evident for the 50% PANI composites.

#### 3.1.3. Fourier Transform Infrared (FTIR) Spectroscopy

Figure 5 displays normalized FTIR spectra of the starting materials. All band assignments were correlated with the literature and are in good agreement with previous results for the composites containing CHT and PANI [4,40,41]. The most noticeable changes occurred at 1509 and 1592 cm^−1^, which correspond to benzenoid (B) and Q-ring stretching of PANI, respectively. In Figure 6, the B and Q bands of PANI changed their relative intensities, especially for 50% PANI in situ composite, where the intensity of the Q ring exceeds the B ring band. A prior report [4] postulates that greater numbers of Q rings enhance the formation of robust and linear structures for PANI/CHT Type 1 composites that reach a maximum at the 50/50 wt.% ratio [4]. The effect of changing the B:Q ratio (in favour of the latter) for PANI upon addition of CHT has a greater stabilization effect on the PANI-CHT interactions.

In Figure 6, the PANI/CHT binary composites showed variable spectra for different PANI fractions of 25%, 50% and 75%. The in situ polymerized samples have lower IR band intensity at 1034–1075 cm^−1^ (C–O stretching) that suggest stabilizing interactions between the –CH_2_–OH groups of CHT and the –NH– groups of PANI. The IR bands at 1312–1318 cm^−1^ (C–N stretching of amide) have lower intensities than those for Type 2 and 3 systems, which implies that –CH–NH– groups of CHT undergo covalent bonding with the PANI moiety. Aside from secondary amine groups, primary –NH_2_ groups of CHT can form Schiff base adducts with PANI. This is shown in the inset (*cf.* Figure 6c), where the intensity of primary –NH_2_ groups (~3530 cm^−1^) decreases with a concomitant increase in intensity of secondary –NH– groups (~3310 cm^−1^). The peak intensity at 1665 cm^−1^ (C=O amide stretching) of PANI gradually decreases as the PANI content varies from 25% to 75%. This suggests that –NHCOCH_3_ functional groups can provide stable chemical interactions with PANI along with –NH_2_ and –OH groups of CHT. Also, the small IR band at 1756 cm^−^^1^ is conspicuous for 50% PANI and invisible for 25% PANI, revealing the formation of potential ester linkages between PANI and CHT, albeit in small amounts.

In comparison with in situ polymerized (Type 1) samples, the intensities at 1034–1075 cm^−1^ of molecularly associated samples have a similar trend, with higher intensity versus the in situ polymerized samples. H-bonding over covalent bonding was inferred for the molecularly associated (Type 2) samples. Compared to in situ polymerized samples, where the ratio of B- to Q-rings of PANI is the opposite (B- over Q-forms of PANI is favoured) for Type 2 samples. The peak intensities at 1663 cm^−1^ (C=O stretching of amide) reveal similar trends to Type 1 samples. At higher fractions of PANI, this peak shifts to 1673 cm^−1^, whereas ester formation (1756 cm^−1^) was not observed for Type 2 samples.

FTIR spectra of physical mixtures (Type 3) differ from both Type 1 and 2 samples. Peak intensity changes were not found at 1150 cm^−1^ (C–O–C bridge of CHT) and 1160 cm^−1^ (B–NH–B linkage of PANI), as compared to the spectra of the PANI and CHT precursors. This trend indicates the absence of any significant bonding interactions for Type 3 powdered solids. The relationship between B- and Q-forms of PANI in physical blends resembles the spectra for Type 2 samples, which indicates that noncovalent interactions prevail for the Type 2 and 3 composites.

In summary, the ^13^C NMR and FTIR data suggest that 25% PANI for Type 1 samples undergo unique chemical bonding. Additionally, 25% and 50% PANI composites possess more rigid linear conjugated structures with greater contribution of Q-forms, whereas 75% PANI composites are more flexible and contain more abundant B-forms of PANI [3].

#### 3.1.4. Raman Spectroscopy

Figure 7a,b shows Raman spectra of the precursor materials, where characteristic spectral bands are noted at 1165 cm^−^^1^ for PANI and 1089–1111 cm^−1^ for CHT, which are in good agreement with the literature [42,43]. In Figure 7c–e, PANI Raman bands are mainly present for the PANI/CHT composites: 1158 cm^−^^1^ (Q ring, C–H bending) and 1496–1595 cm^−1^ (Q ring, C=C stretching). Raman spectral bands of CHT are poorly discernible due to two effects, fluorescence background emission or spectral overlap with PANI bands. Notably, only the 75% PANI (Type 1) composite has a band at 1621 cm^−1^ (C=O vibrations of a benzoquinone fragment). This is an intrinsic spectral characteristic of pristine PANI (benzoquinone fragments, should not be confused with Q-forms of PANI, which are formed as side products from the oxidative polymerization of aniline [44]).

The peak intensities at 1095–1099 cm^−1^ for Type 1 samples gradually fade as the PANI wt.% increases. However, the same bands for Type 2 samples (shifted to 1107–1111 cm^−1^) and remain unchanged. This implies the possible chemical interactions between CHT and PANI in the case of Type 1 composites, whereas such interactions are unlikely for Type 2 materials. Further evidence of covalent bonding between -OH groups of CHT and NH-fragments of PANI may occur by Type 1 systems, as supported by bending vibrations of the -OH groups of CHT (1460 cm^−1^) that tend to vanish for all Type 1 samples, except for 50% PANI. In addition, large changes occur at 1160–1170 cm^−1^ (*cf.* Figure 7c–e). This trend parallels findings derived from solids NMR (Section 3.1.2) and FTIR (Section 3.1.3) spectral results.

In summary, the spectral regions that underwent key changes were ~1100, 1160, 1460 and 1500–1590 cm^−1^. Raman spectral results confirm covalent bonding via –OH and –NH–/–NH_2_ groups for Type 1 samples, but not for samples obtained via the Type 2 method. Type 1 and 2 samples did not show significant wavenumber shifts and band intensities in the Raman spectra versus the Type 1 samples.

### 3.2. Physical Characterization

#### 3.2.1. Thermogravimetric Analysis (TGA)

Wilson and Xue [45] revealed the role of cross-linking of CHT at variable levels of glutaraldehyde, including coordination with Cu(II) due to temperature shifts in the thermogravimetry (TG) profiles versus pristine CHT. Herein, TG was used to provide insight on the composition of cross-linked biopolymers and their thermal stability due to changes in the adhesive interactions upon synthetic modification. TG profiles of single components (CHT and PANI) are shown in Figure 8a. For CHT, the TG profile has a single characteristic thermal event at 294 °C, which corresponds to biopolymer decomposition [23]. By comparison, PANI does not undergo any notable thermal decomposition below 490 °C, whereas the onset of the first TG event begins at 491 °C in Figure 8b, in agreement with another report [4]. PANI, CHT and their binary composites may contain bound water as depicted by thermal events at ~55 °C.

Type 1 biocomposites display features characteristic of densely packed structures (Section 3.1.2 and Section 3.2.2), which concur with the formation of stable complexes, where the PANI chains are inferred to undergo enclathration by CHT chains to form structures that resemble supramolecular helicates [46]. The unique structural features of Type 1 composites are consistent with the formation of covalent grafts between CHT and PANI and/or its ANI oligomers, along with HB formation between the D-A groups of PANI and CHT. The formation of such supramolecular helicates (clathrates) provides an account for the observed stability displayed by Type 1 composites and the effect of variable PANI content in these systems.

Building upon the above discussion, for the case of the binary composites, a unique TG profile is observed for 25% PANI for the Type 1 sample (Figure 8b). Its main thermal event at 250–260 °C splits into two events at 253 °C and 262 °C, unlike the noncovalent composites prepared by Routes 2 and 3 (Figure 8b). The trend suggests that two structural forms of CHT may exist at this elevated composition of CHT (75 wt.%). This is in contrast with CHT, where its thermal event occurs at 294 °C (Figure 8a). The doublet feature of the band between 250 and 260 °C in the TG profile may signify that CHT displays variable thermal stability due to the presence of variable structural forms. At this condition, CHT content (75 wt.%) is in excess over PANI (25 wt.%), where the different structural forms may relate to variable levels of bonding between PANI and CHT. Turning to the 50% and 75% Type 1 composites, the higher TG event shifts to ~323 °C and the onset corresponding to PANI decomposition is lowered to ca. 400 °C and 425 °C, respectively. The lower temperature onset for PANI is noted for the Type 2 and 3 composites, especially as the PANI content reaches 50% and higher. This trend is attributed to disruption of favourable pair-wise interactions that otherwise occur for the pristine (bio)polymer (CHT-CHT or PANI-PANI) systems upon formation of a binary composite system. The negative trend in temperature stability is consistent with breakdown of stabilizing effects due to association of dissimilar polymers (CHT-PANI). The lower onset TG event for PANI may reflect the decreased role of stabilizing pair-wise interactions (*π*-*π*, electrostatic, van der Waals, etc.) between similar polymer units as the mole fraction of the composite deviates from unity for each pure component. As well, the decreased role of covalent bonding for Type 1 composites is supported by the spectral results (*cf.* Section 3.1.2, Section 3.1.3, Section 3.1.4 and Section 3.2.3), along with the formation of aniline oligomers, in agreement with a reduction in the molecular weight distribution of PANI. The variation in the TG events for the composites versus the single component precursors (CHT and PANI) provide support that unique bonding occurs for these biocomposites. A comparison of the noncovalent composites via Routes 2 and 3 (cf. Figure 2) reveal minor differences in the TG profiles. This trend suggests the role of noncovalent association (via HB and van der Waals interactions) in the composites, which parallel the pristine polymers (CHT and PANI). An interesting feature for the 50% PANI Type 1 sample reveals that the occluded water desorbs at much higher temperature (86 °C, that is ca. 30 °C higher versus other samples), providing support for the role of water occluded within pores, as compared with surface-bound water that desorbs ca. 55–60 °C (*cf.* Figure 8a). For 25% PANI, this event occurs over a wider range in the TG profile with the median at 143 °C, revealing stabilizing effects between H_2_O and the composite, which is attributed to the role of pore domains with variable size. In addition, Type 1 composite (25% PANI) systems may possess more micropore sites versus polymer surface sites, which favour bound water. The presence of imine sites for the Q-form of PANI for Type 1 composites was reported to increase for composites with greater wt.% content of CHT [3].

The trends can be briefly summarized by noting that shifts in the TG onset are greater for Type 1 versus Type 2–3 composites. The trends for Type 1 systems are consistent with the formation of covalent bonds, which also include favourable electrostatic interactions between PANI and CHT. By contrast, the trends for Type 2–3 composites are accounted for on the basis of linear combinations of the mixed properties of the single components (CHT, ANI, PANI), as shown in panels (a,b) of Figure 8, which are weighted by their wt.% content. Type 1 composites reveal unique structure and bonding, especially for 25% PANI that displays two structural forms of CHT and 50/75% PANI evidence for oligomers of ANI and PANI grafted onto CHT. The earlier TG onset events for PANI suggest that PANI chains in 50%/75% PANI composites are shorter than for 25% PANI, in agreement with the role of covalent grafting and HB effects between CHT and PANI.

#### 3.2.2. Scanning Electron Microscopy (SEM)

Figure 9 displays SEM images for Type 1–3 composites. The images for Type 1 systems are shown in Figure 9a,b for variable composition (25% and 50% PANI). Figure 9c,d illustrate the SEM images for Type 2 and 3 composites (50 wt.% PANI). All powders were ground with a mortar and pestle in a similar manner, as evidenced by powder grains of variable sizes. A comparison of the composites reveals that the Type 1 samples (Figure 9a,b) possess dense structures with sharp edges and smooth surfaces without any clear indication of macroporosity. By contrast, the molecularly associated (Figure 9c) and physically blended composites (Figure 9d) show increasingly rougher surfaces without clear evidence of macroporosity. The lower density of such Type 2 and 3 composites is indicative of the role of structural defects in the materials or reduced adhesion between the components, which concur with the primary role of weak noncovalent interactions for these noncovalent composites. The reduced formation of large particle grains in Figure 9d is understood due to the blending of CHT and PANI without solvent (Type 3), in contrast to Type 1 and 2 materials, where the solvent employed during preparation facilitates interaction between CHT and PANI.

Type 2 composites have irregular surface edges and appear more amorphous over Type 1 composites. Type 3 systems are highly amorphous with limited evidence of crystallinity, which resemble Type 2 composites. In summary, the particle density of the composites is listed in descending order: Type 1 > Type 2 > Type 3. This trend provides evidence that Type 1 composites have a unique morphology that is distinct from Type 2 and 3 composites. The role of covalent/noncovalent interactions between PANI and CHT for Type 1 systems versus predominantly noncovalent interactions for Type 2 and 3 systems is supported by the TGA results above.

#### 3.2.3. Powder X-ray Diffraction (PXRD)

According to Figure 10a and the literature, both PANI and CHT exhibit two characteristic signatures at 2θ near 9.5° and 20°, respectively [22,47]. The network structure of CHT has prominent crystalline features versus PANI, as observed by the greater band broadening for the PANI XRD lines.

Type 1, 2, and 3 samples show similar trends in spectral broadening: the higher PANI wt.% reveals greater band broadening, indicative of its lower crystallinity. PANI 25% prepared via Route 1 vs. Route 2 or 3 reveal the most strikingly different 2θ values at 19.9° and 20.6°, respectively. The composites with 50% and 75% PANI, as well as Type 2 and 3 samples, maintain similar 2θ values. This trend indicates that unique spectral signatures occur, which may support that chemical interactions for PANI 25% appear for Type 1 samples that correlate favourably with other spectral data (Section 3.1.2, Section 3.1.3 and Section 3.1.4). Covalent bonding between CHT and PANI may enhance the crystallinity of a composite by enhancing the noncovalent association between the (bio)polymer units, as reflected by the sharper XRD bands and higher structural order. The other minor reflections (ca. 38°, 44° and 64°) are visible mainly for physically blended composites (Figure 10d). The XRD bands relate to the pristine components (Figure 10a) or due to the residual inorganic salts, such as sulphates, as the reduction products of a persulfate anion that undergo occlusion within the crystals or at the protonated imine sites of PANI.

#### 3.2.4. Solvent Swelling Tests

Water uptake by biopolymers was shown to provide insight on the role of accessible hydrophilic sites and features of the textural properties [45], as evidenced for the hydration properties of starch and cellulose [48]. Herein, the equilibrium solvent swelling results of the Type 1–3 composites, CHT, and PANI were studied to gain insight on the role of structural variations of the composites prepared by Routes 1–3. According to Table 2, the magnitude of the solvent swelling results in water for pristine polymers (PANI (400%) and CHT (350%)) are relatively high and comparable in magnitude, which agree with reported values for CHT, 357% [49] and PANI, up to ca. 400% [50]. It is noteworthy that some variability in swelling can be accounted for according to the charge state of the amine/imine groups of CHT and PANI in their protonated forms (Figure 4b). The variable hydrophile-lipophile balance (HLB) of CHT and PANI and the accessibility and abundance of polar groups will affect the resulting swelling of the binary composites, as revealed in Table 2.

In Table 2, the magnitude of the swelling value is listed in descending order: Type 3 > Type 2 > Type 1, where the trend can be related to the nature of the interactions between the PANI/CHT composites, covalent versus noncovalent interactions. The lowest swelling occurs for Type 1, which may relate to the combined effects of strong adhesion (covalent + noncovalent interactions) between PANI and CHT. The greatest swelling occurs for Type 3 systems, whereas Type 2 systems have intermediate swelling, which concur with role of noncovalent interactions that are mediated by the solvent. Type 3 systems likely have few net intermolecular (noncovalent) interactions between PANI and CHT since the preparation occurred in the absence of solvent. Type 2 composites have attenuated association between the (bio)polymer chains by breakage of H-bonds between the D-A groups, as described in a recent review [2]. Trends in swelling according to association between D-A complexes are supported by the results reported for polyelectrolyte complexes of CHT [51]. In turn, adhesive interactions affect the packing density of the composites, which is supported by the trends in particle morphology, as evidenced by the SEM results (*cf.* Figure 9), along with variation in the XRD profiles (*cf.* Figure 10) that indicate the relative crystallinity of the PANI-CHT systems. This trend follows the variable degree of association of the polymer subunits that have variable accessibility of hydrophilic functional groups of CHT, as supported by the results reported for polyelectrolyte complexes of CHT [51]. Type 1–3 composites illustrate variable swelling, where reduced swelling values indicate reduced accessibility of hydrophilic groups, which concur with enhanced bonding between the D-A groups of PANI and CHT (Figure 1b). Type 1 materials undergo unique adhesive interactions, attributed to the role of covalent bonding due to cross-linking, along with noncovalent interactions between PANI and CHT (*cf.* Figure 1a). For the case of Type 3 materials (physical mixtures), these systems consist of random assemblies (physical mixtures) of powder grains, where the swelling values can be estimated using an additivity swelling scheme (see footnote in Table 2) based on the single component swelling parameters (especially for a 25% PANI sample). Type 3 systems are unlikely to possess any unique structural attributes since the Type 3 materials are akin to random physical mixtures prepared without solvent (*cf.* Figure 2). By contrast, Type 1 composites display highly unique swelling properties that are not accounted for by the additivity scheme, whereas Type 2 and 3 systems are adequately accounted for by the additivity scheme. Type 1 composites possess much lower solvent swelling (60–155%), whereas Type 2 composites have intermediate swelling (180–280%). By comparison, the highest swelling values (380–470%) occur for the Type 3 composites. This trend concurs with the solvent swelling values (346–398%) obtained for the single component (bio)polymers in proportion to their relative composition. In all cases, similar trends are noted, where greater swelling occurs at higher fractional PANI content. The higher swelling of pristine PANI over CHT can be accounted for by the greater surface charge of PANI in its quinoid (Q) form over a wider pH range [3]. PANI has a higher pK_a_ value (~9) compared to CHT (~6.5), which results in a higher proportion of charged (imine) sites of PANI [52,53]. The trends in Table 2 are complemented by the spectral results in Section 3.1.2, Section 3.1.3, Section 3.1.4 and Section 3.2.3, along with the TGA and SEM results (Section 3.2.1 and Section 3.2.2), which support that covalent and noncovalent bonding occurs for Type 1 composites.

#### 3.2.5. Equilibrium Dye Adsorption

Methylene blue (MB) is a cationic dye that has versatile utility as a probe for the study of CHT-based composites [4,54,55]. Herein, the structural attributes of Type 1–3 composites were studied according to the adsorption profiles of MB at equilibrium conditions. The monolayer uptake capacity ($Q_{m}$) for the composites was estimated by the Sips model [56]. An alternative model for estimating the adsorption parameters involves the use of the Dubinin–Astakhov (DA) equation [24,57]. The DA model affords estimation of the adsorption capacity and the effective adsorption energy, which is unique for a specific adsorbent–adsorbate system. The adsorption isotherm profiles for the various composites are plotted as the adsorbed amount of MB $\left(Q_{e}\right)$ versus the residual MB dye concentration ${(C}_{e}$) in aqueous media (Figure 11).

The composites (Type 1–3) show a nonlinear increase in $Q_{e}$ with increasing $C_{e}$, where unique dye adsorption is evident for the Type 1 versus the Type 2 and 3 composites. Type 1 composites (Figure 11b) display incremental MB uptake with greater CHT content (%), where the adsorption capacity is ca. 10-fold higher than the Type 2 and 3 composites (Figure 11c,d). A comparison of the profiles for Type 2 (Figure 11c) and Type 3 (Figure 11d) composites reveals subtle differences according to their relative PANI/CHT content. Evidence of the unique structure for the Type 2 and 3 composites is revealed by the distinguishable dye uptake results. The Type 2 composites with 25% and 50% PANI (Figure 11c) are lower than the 75% PANI, which may indicate reduced functional group accessibility for PANI as the CHT content exceeds the 1:1 ratio (50 wt.%). The uptake profile for 75% PANI (Figure 11c) converges with pristine PANI (Figure 11a). This indicates that the aniline units of PANI have greater affinity with MB over that of the glucosamine units of CHT, in agreement with differences in their basicity. The closely overlapping profiles in Figure 11d indicate that the maximum uptake for the physical mixture can be modelled by a linear combination of the respective $Q_{m}$ values for PANI (4.99 µmol/g) and CHT (2.82 µmol/g) in a similar manner as the solvent swelling (*cf.* Table 2).

According to the best-fit results for the Sips and DA models, the $Q_{m}$ values obtained by both isotherm models show convergent estimates within ca. 10%. The corresponding model parameters are listed in Table 3, which include the effective binding energies of MB with the various composite adsorbents.

Type 1 composites display the highest adsorption capacities, up to 10-fold higher versus Type 2 composites. The markedly greater MB uptake for the Type 1 composites strongly supports the unique structural features relative to the Type 2 composites. Irrespective of the potential role of bonding for the Type 1 and 2 PANI/CHT composites, the in situ polymerized composites are notably more stable, whereas the molecularly associated (Type 2) composites show considerable variation in dye uptake according to their composition. The active adsorption sites for Type 2 composites are attenuated due to H-bonding between CHT and PANI as the content of CHT increases above 1:1 (>50 wt.%) due to effective shielding of the base sites of PANI. This compositional dependence may relate to the structural role of CHT to form supramolecular helicates as the content of CHT reaches 50% or higher (e.g., 25 wt.% PANI). This trend follows the apparently higher stability (according to water swelling) for Type 1 composites prepared by Route 1 [58].

A comparison of the MB dye adsorption results with the literature in Table 3 reveals that Type 1 composites (unspecified ratios) have a $Q_{m}$ value for MB of 15.4 µmol/g [4], which falls in a range between 10.5 and 59.3 µmol/g, as listed in Table 3. The adsorption capacity of pristine CHT (2.69 µmol/g) concurs with the literature values in Table 3 [13]. The dearth of information about DA parameters for PANI/CHT systems reveals that the reported binding energy for PANI–Congo Red (7.8 kJ/mol) [59] is slightly higher than that for the MB results herein (4.8 kJ/mol for PANI), which may relate to the differences in the pH conditions and charge state since Congo Red is an anionic dye with a higher molecular weight than MB. As such, anionic dyes should bind more strongly with PANI in its protonated state over cationic dyes such as MB due to repulsive charge effects. However, PANI can exist in variably acid-doped states, according to the relative proportion of B and Q forms of PANI. Similar binding energy values may also reveal the role of π-π* interactions between the arene units of the dye and the arene units of PANI. The binding energy for CHT was lower than PANI, which is comparable and within 36% (0.7 vs. 1.1 kJ/mol) based on recent literature data [60]. The trend is consistent with the offset in $Q_{m}$ values (within 10%) for CHT and PANI, respectively.

In Table 3, the adsorption interaction energies for PANI/CHT composites vary from 1.3 to 4.2 kJ/mol, which is higher than the interaction energy of pure CHT (1.1 kJ/mol) and lower than that of pristine PANI (4.8 kJ/mol). This indicates that PANI and CHT contribute a proportional number of accessible adsorption sites in these systems. CHT is non-aromatic and contains –OH groups, where –OH-π interactions are known to provide some stabilized binding in macromolecular systems [61]. In some cases, composites with higher $Q_{m}$ values have greater binding energies. In a report on sulfonated CHT [62], stronger binding with MB was driven by negatively charged sulfonate moieties, where the difference in binding energy was more than 1 order of magnitude (21.8 kJ/mol vs. 1.14 kJ/mol) versus pristine CHT. Another report on phosphorylated CHT shows even stronger binding with anionic Acid Red 88 dye, which occurs due to electrostatic interactions and also at a cost of n-π and Yoshida H-bonding (89.0 kJ/mol) [61]. In Table 3, it is noteworthy that the composite with 50% PANI has a unique E_a_ value relative to 25% and 75% PANI. The 50% PANI composite is inferred to yield a more stable helicate structure when the wt.% ratios are stoichiometric (ca. 50%). The offset in $Q_{m}$ values for MB is 44% lower for CHT compared to PANI, in accordance with differences in their relative basicity.

Another important quantity in Table 3 is the heterogeneity parameter (*n_s_*). The Sips model provides limited insight on the heterogeneity factor,$\;n_{s}$. Hence, the data in Table 3 are shown for the DA model. The heterogeneity factor for many adsorbents (such as activated carbons, clays and resins) vary from 1 to 3, whereas zeolites exhibit higher *n_s_* values (typically > 4) [63]. An adsorbent has a uniform pore structure when the heterogeneity factor ≈ 2. For this condition (*n_s_* = 2), the DA model transforms into Equation (6), initially postulated by Dubinin and Radushkevich [18]. As $n_{s}$ converges to 1, it yields the Freundlich equation [43].

The adsorbent-MB system is most favourably described by the Dubinin–Radushkevich (DR) model occurs for 50% PANI for the Type 1composite, *n_s_* = 2.05. This suggests that 50% PANI content yields a pore geometry and distribution that is more uniform. For 25% PANI (Type 1) and 75% PANI (Type 2), the heterogeneity parameter is nearly 1, which agrees with the Freundlich model, and is valid for non-uniform pore structures and heterogeneous surface sites. This trend provides support for the formation of supramolecular helicates, as indicated above. By comparison, the 25% and 75% systems have a non-uniform pore structure due to defects upon departure from the 50% wt. ratio of PANI. Remarkably, physically mixed samples (50% and 75% PANI Type 3) are similar to pristine PANI in terms of heterogeneity (*n_s_*~1.5), whereas 25% PANI (Type 3) sample resembles pristine CHT (*n_s_*~2.3–2.6). This trend provides evidence of the absence of chemical interactions for PANI and CHT, which contribute proportionally to the adsorption properties and internal pore structure of a composite. Hence, the adsorption properties can be accounted for using a linear additivity scheme similar to that reported for the swelling results in Table 2 and elsewhere [13].

Building upon the solvent swelling discussed in Section 3.2.4, the surface chemical features for Type 1 to 3 composites were studied using MB as a dye probe to evaluate differences in structure of the materials [4,54]. Figure 11 illustrates MB dye uptake results for the composites versus the residual MB concentration (C_e_), along with the adsorbent-dye binding energy (ε), obtained by the Dubinin-Astakhov model. A visible difference in MB uptake (ca. 10-fold) is noted between Type 1 versus Type 2 and 3 composites in Figure 11a–d. The increase in adsorption capacity relates to the nature of the adsorption sites and their accessibility. The greater MB dye uptake for Type 1 composites reflects the role of covalent cross-linking plus non-covalent stabilizing interactions. This is in contrast with noncovalent bonding for Type 2 and 3 systems, where the experimental details and results of the isotherms are further described below.

In summary, the use of MB as the dye probe for adsorption supports that Type 1 composites (50% PANI) had the highest adsorption capacity. In combination with the spectral data outlined above, the swelling tests, among the other results, provide strong support for the unique structural contributions that arise from (non)covalent bonding between CHT and PANI, which occur for Type 1 composites. H-bonding interactions among D-A sites of each (bio)polymer subunit between the complementary functional groups of CHT and PANI. Type 2 samples do not show a significant difference from physical blends in terms of their adsorption properties. Once immersed in solution, the HB complex may become partially or fully dissociated depending on the extent of water competition with the corresponding D-A sites of CHT and PANI. Table 3 lists the best-fit parameters for the MB uptake capacities and adsorbent-to-dye binding energies, which were estimated according to the Sips and the D-A isotherm models, respectively. The results in Table 3 generally support that the composites display adsorption sites with variable heterogeneity (*n_s_* = 0.5–2.3) and variable adsorbent-dye binding energy (1.6–4.2 kJ/mol), according to the composition of the system, which are well-described by the isotherm models (*cf.* Equations (5) and (8)).

#### 3.2.6. Solubility Tests

Table 4 lists the solubility of PANI/CHT Type 1 composites, PANI, and CHT in dimethyl sulfoxide (DMSO). It is also known that PANI can form complexes with water and organic solvents, where each repeat unit of PANI can bind with a few solvent molecules [64]. PANI has a limited solubility in DMSO in its EB form, whereas CHT is nearly insoluble, if not converted into CHT-acid salts [65]. This correlates well with the data listed in Table 4.

Among the various systems in Table 4, PANI has the highest overall solubility. As the CHT/PANI wt.% ratio increases, the solubility of the composite drops due to the low solubility of CHT. On the other hand, the most CHT-rich composite (25% PANI) does not exhibit the lowest solubility; this may be attributed to the specific non-covalent interactions, which can be overcome upon intrusion of the solvent. The sample containing 50% PANI shows the unique and lowest solubility in DMSO among the three binary composites. Supporting evidence of stronger covalent bonding and/or stable noncovalent interactions concur with the SEM results in Figure 9. Based on the trend in solubility for the binary composites, CHT is inferred to enclathrate PANI, where the outer surface of the composite portion in contact with the solvent is mainly CHT, whereas PANI is included within the CHT biopolymer framework. The stoichiometric wt. ratio of 50% aligns with the solubility trend in Table 4. The 75% PANI refers to an excess level of PANI relative to CHT, beyond the ideal 50 wt.% PANI, where the excess PANI is inferred to impart enhanced solvation and overall solubility to the composite. This concurs with the solvation of 75% PANI, with as many as five DMSO molecules for each aniline unit. The enhanced solubility of 75% PANI is consistent with a change in polarity of the composite relative to 50 wt.% PANI, which provides compelling evidence in support of the supramolecular helicate (clathrate) structure, as described above.

### 3.3. Chemistry of Formation of PANI/CHT Type 1 Composite

Building upon the results presented in Section 3.1.1, there are two carbon sites of CHT that undergo chemical change (C6- and C2-). The hydroxymethyl (–CH_2_OH) group of CHT represents a primary alcohol that is susceptible to oxidation. APS is a strong oxidizer that transforms a primary alcohol into an aldehyde under the employed acidic conditions (HCl)_aq_. Further, the aldehyde reacts with –NH_2_ groups of PANI to form a Schiff base (*cf.* Figure 12a). Another site for oxidation of CHT is C2, since it can undergo a condensation reaction (*cf.* Figure 12b).

## 4. Conclusions

This study investigates the bonding interactions and structural properties of polyaniline (PANI) and chitosan (CHT) biocomposites synthesized via three distinct routes: in situ polymerization (Route 1), molecular association (Route 2), and solid-phase physical mixing (Route 3). Our findings demonstrate that Route 1 yields composites with unique structural characteristics and dual bonding contributions (covalent and noncovalent), whereas Routes 2 and 3 primarily exhibit noncovalent bonding. The stability of composites prepared by Route 2 exceeds that obtained for Route 3 due to the role of solvation-assisted complexes between CHT and PANI by Route 2. The composites prepared by Route 3 are largely physical mixtures with apparently negligible formation of stable complexes between CHT and PANI.

The distinct properties of Type 1 composites include significantly reduced solvent swelling and up to a ten-fold increase in dye uptake, as compared with Type 2 and 3 composites that display variable noncovalent interactions. The role of covalent bonding is a distinguishing feature that governs the structure-property relationships of Type 1 systems. Spectral techniques (NMR, FTIR, and Raman) support the variable structure of the composites, along with PXRD, SEM, and TGA results for each type of composite. These insights suggest that the dual bonding (covalent and noncovalent) for Type 1 composites contributes to their unique structural features, along with the trends in the solvation and dye adsorption results that confer distinguishable surface chemical properties.

The results have significant implications for the tailored design of PANI/CHT composites for advanced applications, including functional adsorbents, redox-active catalysts, and electrochemical sensors [66]. Future research directed toward optimizing the synthesis conditions to tailor the supramolecular organization between CHT and PANI will yield improved physicochemical properties. In turn, studies of the functional properties of PANI/CHT composites and their utility in diverse technological applications offer promising lines of future investigation.

## Figures and Tables

**Figure 1 polymers-16-02663-f001:**
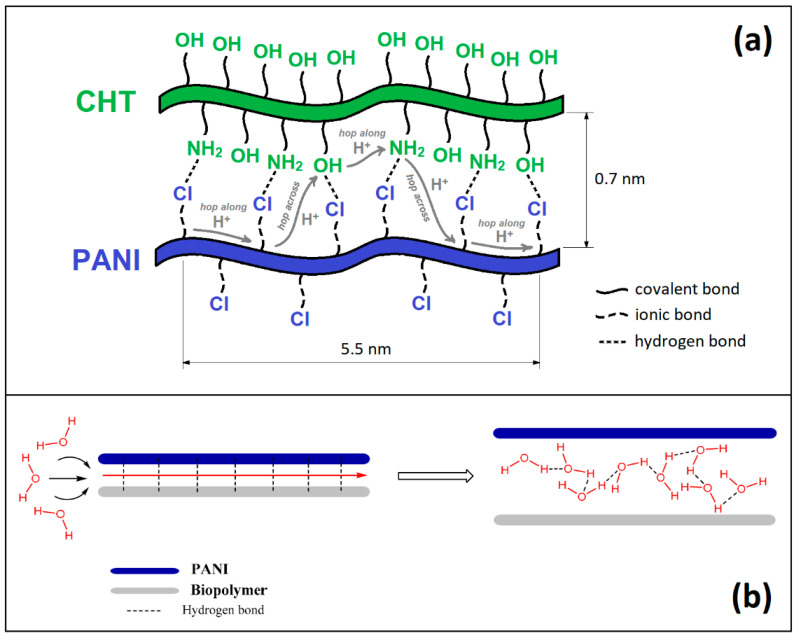
(**a**) Proton-hopping mechanism in PANI/CHT composites; (**b**) destruction of internal hydrogen bonds in a composite at elevated humidity, followed by dissociation of polymer chains, where H_2_O molecules serve as carriers of electrical charge. Reprinted with permission from [2].

**Figure 2 polymers-16-02663-f002:**
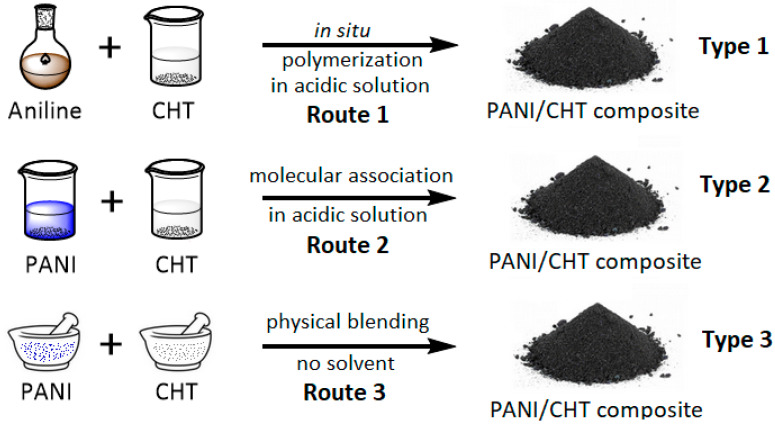
Preparative routes (1–3) for synthesis of PANI/CHT composites (Types 1–3).

**Figure 3 polymers-16-02663-f003:**
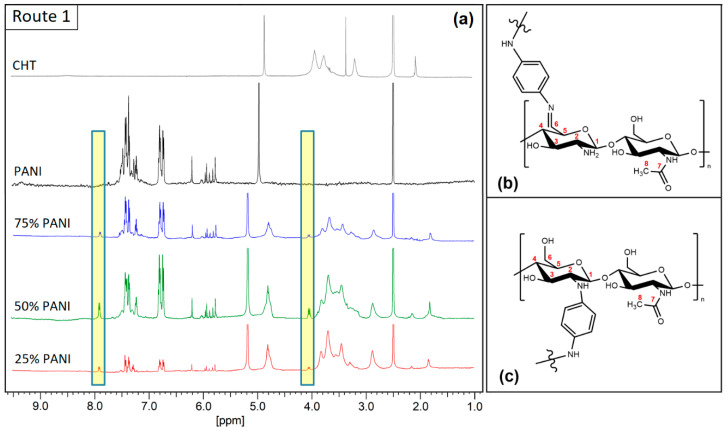
(**a**) ^1^H NMR spectra of pristine (bio)polymers (**top**) and Type 1 polymer composites (**bottom**) recorded at 295 K and 500 MHz in DMSO-*d*_6_/D_2_O/LiCl. Chemical shifts are referenced relative to TMS (0.0 ppm). Schematic structures of CHT-PANI crosslinking: (**b**) C6 of –CH_2_OH group; and (**c**) C2 of the glucopyranose ring of CHT.

**Figure 4 polymers-16-02663-f004:**
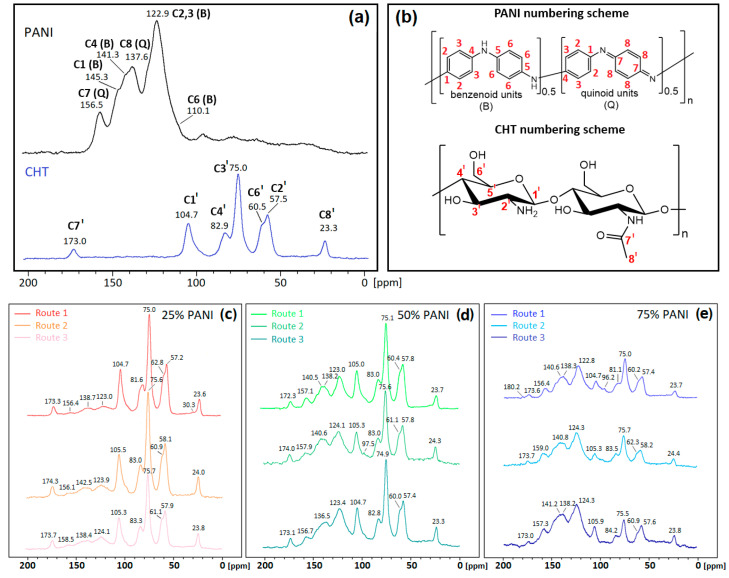
Solid-state CP-TOSS ^13^C NMR spectra obtained at 295 K with a TOSS rotational speed (7.5 kHz), recycle delay (2 s), and cross-polarization time (1 ms): (**a**) precursor materials; (**b**) C-atom numbering scheme for PANI and CHT; (**c**) 25 wt.% PANI; (**d**) 50 wt.% PANI; and (**e**) 75 wt.% PANI. Panels (**c**–**e**) present normalized spectra for each panel.

**Figure 5 polymers-16-02663-f005:**
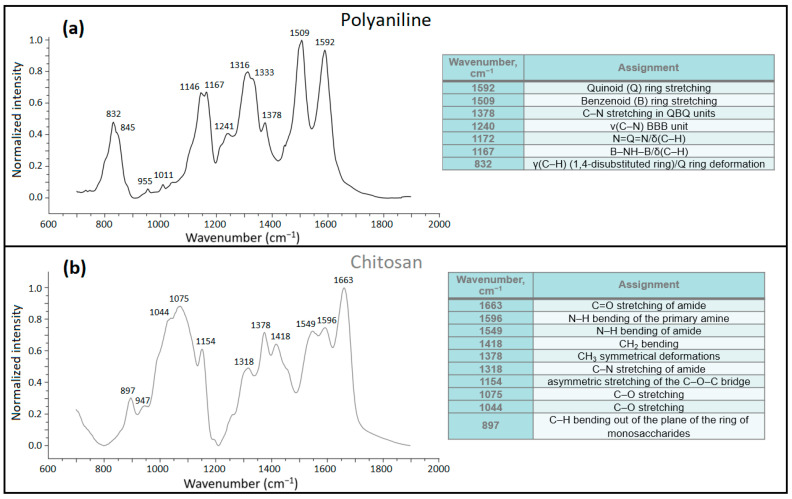
Normalized FTIR spectra of pristine materials: (**a**) PANI and (**b**) CHT. Spectral regions between 1900 and 3500 cm^−1^ were omitted for clarity.

**Figure 6 polymers-16-02663-f006:**
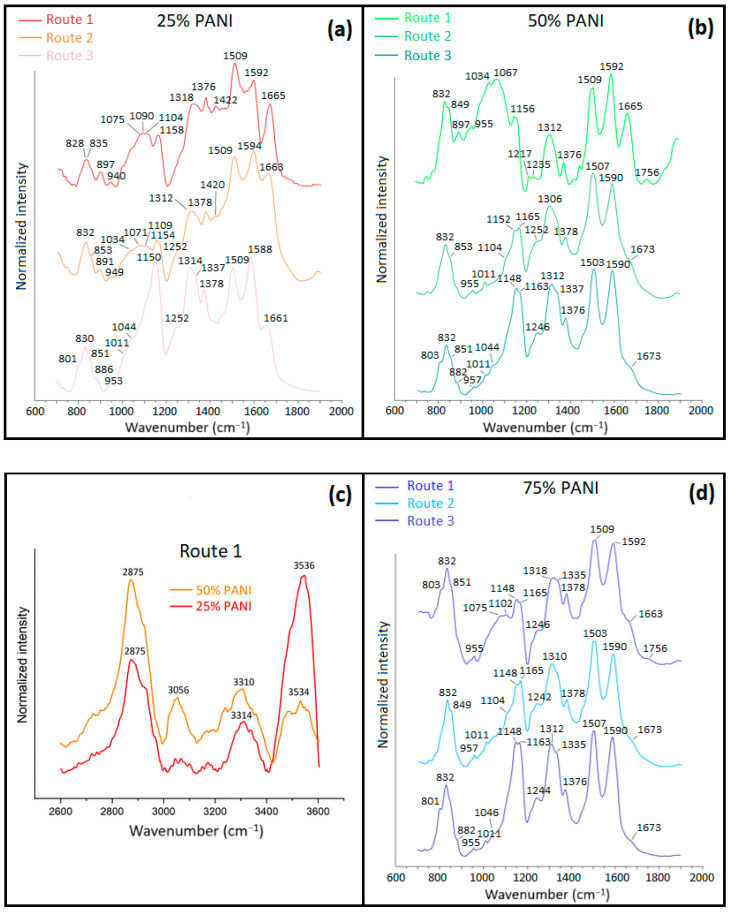
Normalized FTIR spectra of composite PANI/CHT materials: (**a**) 25% PANI, (**b**) 50% PANI, and (**d**) 75% PANI in the sequence of in situ polymerized—molecularly associated—physically blended samples, respectively. Spectral regions between 1900 and 3500 cm^−1^ were omitted for clarity. Panel (**c**) shows the inset of the region 2600–3600 cm^−1^ for 50% and 25% PANI prepared via in situ polymerization (Route 1).

**Figure 7 polymers-16-02663-f007:**
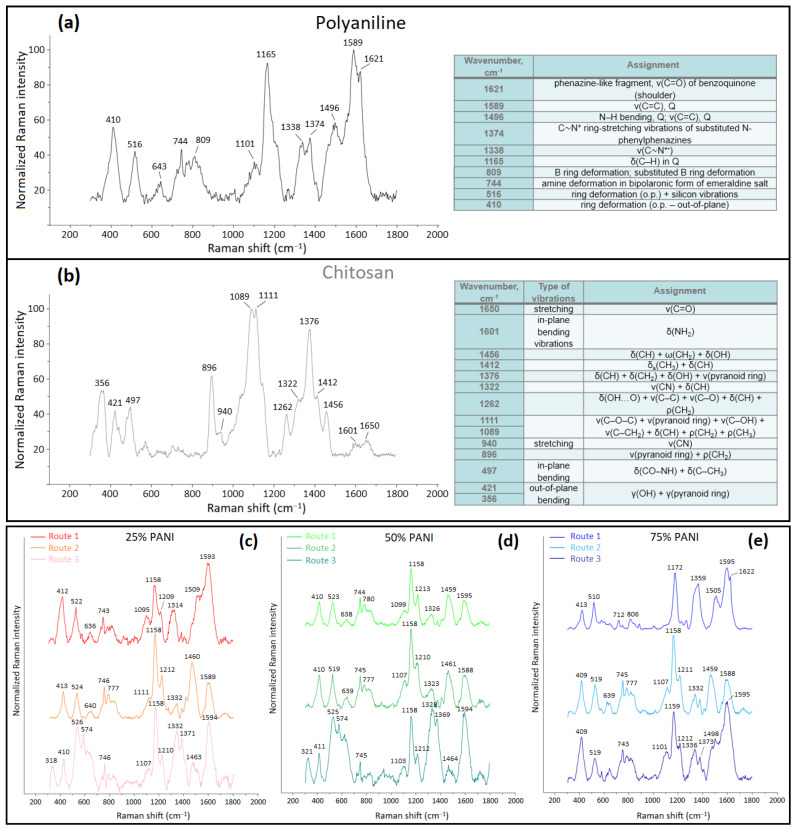
Normalized Raman spectra of pristine materials: (**a**) PANI and (**b**) CHT; and composite PANI/CHT materials: (**c**) 25% PANI, (**d**) 50% PANI and (**e**) 75% PANI in the sequence for in situ polymerized—molecularly associated—physically blended mixtures at $\lambda_{exc}$ = 785 nm.

**Figure 8 polymers-16-02663-f008:**
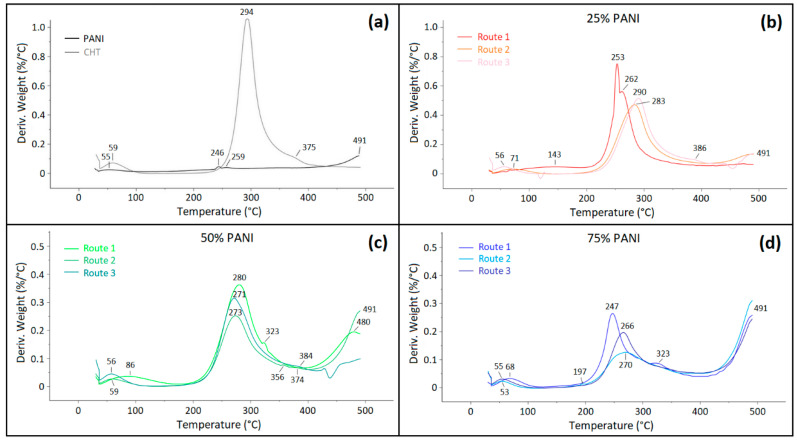
Thermogram profiles of polymers (PANI, CHT) and their binary composites: (**a**) single components: CHT and PANI; and binary composites: (**b**) 25%, (**c**) 50% and (**d**) 75% PANI for composites prepared by Routes 1–3, respectively.

**Figure 9 polymers-16-02663-f009:**
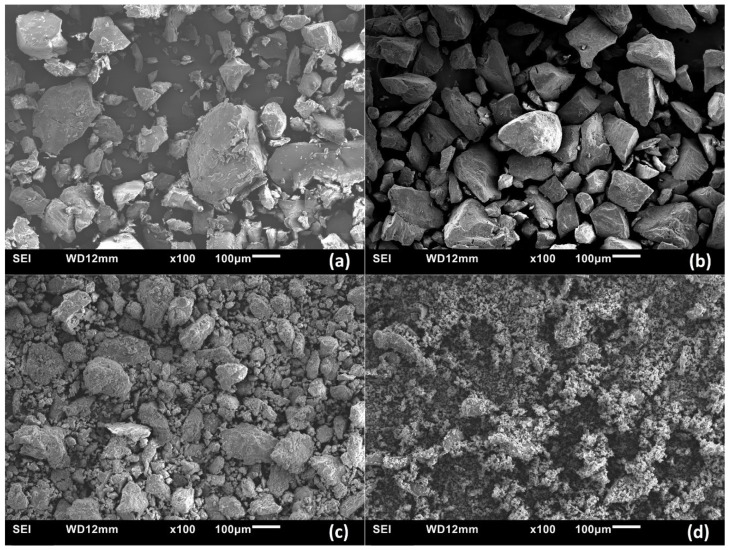
SEM micrographs of (**a**) 25% PANI and (**b**) 50% PANI in situ polymerized samples; (**c**) molecularly associated and (**d**) physical blends at 50% PANI weight fraction.

**Figure 10 polymers-16-02663-f010:**
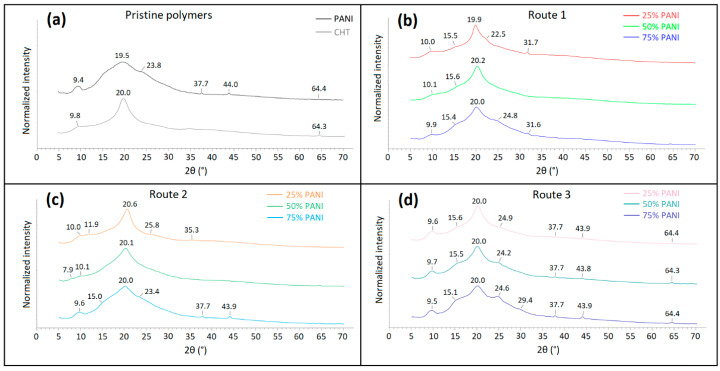
Powder XRD profiles of precursors and composites according to synthetic Routes 1–3: (**a**) pristine CHT and PANI, (**b**) in situ polymerized, (**c**) molecularly associated and (**d**) physically mixed PANI/CHT composites (Cu K-α irradiation source, λ = 0.154 nm).

**Figure 11 polymers-16-02663-f011:**
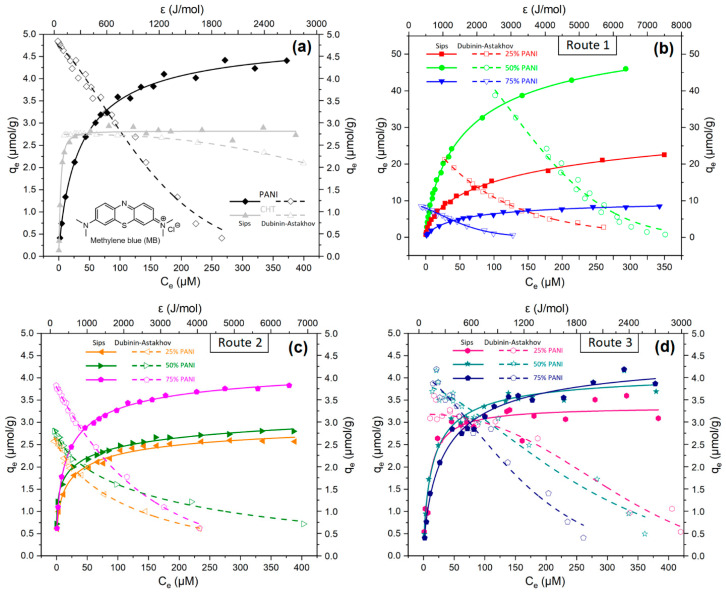
Equilibrium dye uptake of MB at 295 K and pH 7: (**a**) pristine PANI and CHT, (**b**) in situ polymerized (Type 1), (**c**) molecularly associated (Type 2) and (**d**) physically blended (Type 3) PANI/CHT composites.

**Figure 12 polymers-16-02663-f012:**
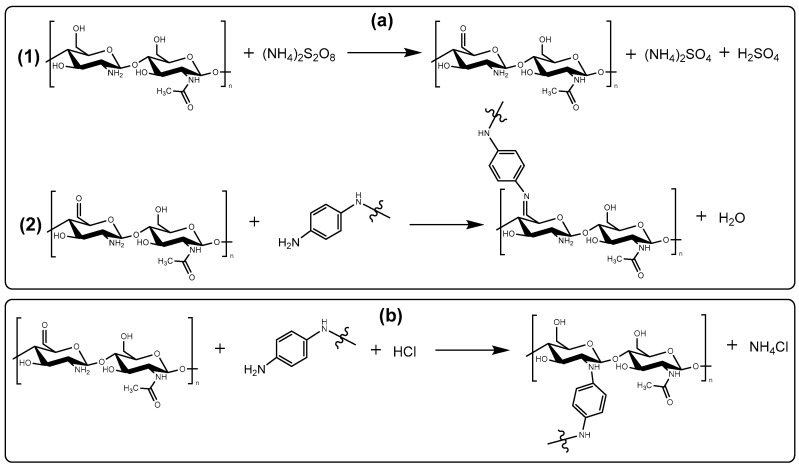
Schematic illustration of (**a**) oxidative addition of PANI and CHT via the C6 position of CHT (Schiff-base formation); and (**b**) polycondensation between PANI and CHT units via C2 position of CHT.

**Table 1 polymers-16-02663-t001:** Characteristic ^1^H NMR chemical shift (δ) values for PANI and CHT.

Assignment	δ, ppm *	Multiplicity	Integration **	Number of H Atoms
**PANI signatures**				
Benzenoid rings (B)	7.2–7.6	multiplet	4.00	4
Quinoid rings (Q)	6.7–6.9	multiplet	5.90	4
**CHT signatures**				
C1 *** proton	4.7–4.9	broad peak	1.14	1
C6 protons	3.8–4.0	broad peak	2.36	2
C4, C5 protons	3.5–3.7	broad peak	1.90	2
C3 proton	3.3	multiplet	0.924	1
C2 proton	3.2	broad peak	1.25	1
C8 (–CH_3_) protons	1.8–1.9	singlet	0.573	3
**Binary composite (50% PANI) signatures ****				
–C6H=N–	7.93	doublet	0.192	1
–C5H–	4.05	multiplet	0.170	1

* Chemical shifts are shown for pristine components relative to TMS (δ = 0), whereas the NMR lines in composites may have variable chemical shift values (*cf.* Figure 3). ** Peak integration and new peaks are shown for stoichiometric PANI/CHT ratio (50% PANI composite). *** For the CHT numbering scheme, refer to Figure 3b,c.

**Table 2 polymers-16-02663-t002:** Water swelling at ambient conditions for PANI, CHT and Type 1–3 composites prepared by the various preparative routes (Routes 1–3; see Figure 2).

Sample Preparation Route	Sample	Experimental Swelling in Water (%)	Theoretical Swelling * (%)
**Type 1 Composite** (In situ polymerization)	25% PANI	60 ± 26	360
	50% PANI	111 ± 31	380
	75% PANI	155 ± 31	390
**Type 2 Composite** (Association in water)	25% PANI	180 ± 9	360
	50% PANI	237 ± 17	380
	75% PANI	284 ± 9	390
**Type 3 Composite** (Physical mixing of solids)	25% PANI	379 ± 30	360
	50% PANI	405 ± 13	380
	75% PANI	422 ± 8	390
Pristine polymer Pristine biopolymer	PANI	398 ± 42	398
	CHT	346 ± 24	346

* $S_{total}=\chi_{CHT}S_{CHT}+\chi_{PANI}S_{PANI}$, where $S_{i}$ represents the solvent swelling of the respective (bio)polymer and $\chi_{i}$ represents the mole fraction of the (bio)polymer.

**Table 3 polymers-16-02663-t003:** Monolayer uptake capacities and adsorbent-dye binding energies obtained at 295 K, according to Sips and Dubinin–Astakhov isotherm models.

Sample Preparation Route	Sample	$Q_{m}$ (µmol/g)		Heterogeneity Parameter	^b^$E_{a}$ (kJ/mol)
		^a^ Sips	^b^ DA	^b^ *n_s_*	
**Type 1 Composite** (in situ polymerization)	25% PANI	39.6	35.2	0.98	2.13
	50% PANI	59.3	57.9	2.05	4.21
	75% PANI	10.5	9.14	1.66	1.61
**Type 2 Composite** (Association in water)	25% PANI	3.23	2.98	0.72	2.17
	50% PANI	4.09	3.37	0.52	3.25
	75% PANI	4.42	4.07	1.01	2.30
**Type 3 Composite** (Physical mixing of solids)	25% PANI	3.35	3.18	2.34	2.43
	50% PANI	4.15	3.86	1.49	2.02
	75% PANI	4.58	4.08	1.48	1.28
Pristine polymer	PANI	4.99	4.77	1.47	4.82
Pristine biopolymer	CHT	2.82	2.75	2.58	1.14

^a^ Sips model (*cf.* Equation (5)) and ^b^ DA model (*cf.* Equation (8)).

**Table 4 polymers-16-02663-t004:** Solubility of PANI/CHT composites in DMSO and their complexes.

Sample	Formula of Complex	Solubility, mg/g
25% PANI (Type 1)	(PANI/CHT)·2⅖DMSO	0.820
50% PANI (Type 1)	(PANI/CHT)·2215DMSO	0.107
75% PANI (Type 1)	(PANI/CHT)·5¾DMSO	1.62
PANI	PANI·2DMSO	2.09
CHT (non-ionized form)	CHT·⅛DMSO	0.00119

## Data Availability

The raw data supporting the conclusions of this article will be made available by the authors upon reasonable request.

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
