# Peer review of "Chitosan-Polyaniline (Bio)Polymer Hybrids by Two Pathways: A Tale of Two Biocomposites"

_polymers, 2024, doi:10.3390/polym16182663_

Round 1

Reviewer 1 Report

Comments and Suggestions for Authors

Dear editor, dear authors,

 This article is a very interesting piece of science that deserves to be published.

The article is well written, systematic and logic and I just profit of this opportunity to congratulate the authors.

A few aspects can be considered to improve the article even further.

-        Please present the chemical characterization with NMR (1H and 13C) and with FT-IR and raman in subsequent paragraph.

-        Please report the granulometry of the polymers for the swelling test ( I guess they should be all the same, but for route 1 and 2 is not specified and the SEM images let me with some doubt in this regard).

-        Details on the amount of reactant and solvent (so as acid and catalyst in case of route 1) should be given in the experimental section.

-        It would be nice to understand the tiny signal at 30.3 ppm in 13C-NMR to which methyl group it would belong. I see it also in the route 2 indeed and I also believe it could be due to new covalent especially if you couple with the information of the vibrational spectroscopy.

-        Overall I am convinced that some small covalent can be obtaining also with the route 2.

-        Final observartion: Try to compress the abstract by keeping just the main information.

Author Response

Author Response to Reviewer Comments on MS ID: polymers-3188843 entitled “Chitosan-Polyaniline (Bio)Polymer Hybrids by Two Pathways: A Tale of Two Biocomposites

The authors have addressed the reviewer comments in a point-by-point fashion, as outlined below in red font. The corresponding edits are reflected in the revised manuscript. We appreciate the insightful and constructive comments provided by the reviewers, along with the opportunity to improve the quality of this work.

Reviewer #1

This article is a very interesting piece of science that deserves to be published.

The article is well written, systematic and logic and I just profit of this opportunity to congratulate the authors.

A few aspects can be considered to improve the article even further.

-        Please present the chemical characterization with NMR (1H and 13C) and with FT-IR and raman in subsequent paragraph.

Response:

To organize the chemical characterization in a logical sequence, which begins with 1H and 13C NMR, followed by FTIR and Raman spectroscopy, a new subsection was added entitled "Chemical characterization" (section 3.1). Therein, each technique is described and its findings are provided in detail, along with consolidation into one coherent section. Other parts were united in another new section "Physical characterization" (3.2). All sections, figures, and tables were renumbered accordingly, as well as any references to them in the accompanying text.

-        Please report the granulometry of the polymers for the swelling test (I guess they should be all the same, but for route 1 and 2 is not specified and the SEM images let me with some doubt in this regard).

Response:

We appreciate the observation regarding the particle size differences as seen in the SEM images. The presence of larger particles in the SEM images for the in situ polymerized samples (Route 1) indeed reflects the distinct mechanical properties of these composites. The larger particle size is attributed to the increased hardness and compactness of the particles formed via the in situ polymerization method. Qualitatively, the particles obtained by Route 1 require notably greater force to break the particles down versus those obtained by Routes 2 & 3. This observation is consistent with the stronger bonds (covalent) formed in Route 1 composites, in accordance with the more robust nature of these particles.

However, for the swelling tests, all samples were thoroughly sieved to ensure particle sizes of 75 µm or smaller. This sieving step was carefully performed to ensure particle size uniformity for all samples, which eliminates any potential bias due to particle size differences for the solvent swelling and adsorption experiments. We added a paragraph to Section 2.5.4 for the solvent swelling tests.

-        Details on the amount of reactant and solvent (so as acid and catalyst in case of route 1) should be given in the experimental section.

Response:

An appropriate paragraph was added to Section 2.4.1.

-        It would be nice to understand the tiny signal at 30.3 ppm in 13C-NMR to which methyl group it would belong. I see it also in the route 2 indeed and I also believe it could be due to new covalent especially if you couple with the information of the vibrational spectroscopy.

Response:

Your observation regarding the small 13C NMR signal at 30.3 ppm noted for the product from Route 2 is greatly appreciated. While this signal may indicate the presence of residual covalent bonds, it is important to note that the line intensity is substantially lower compared to the corresponding 13C NMR signal noted in in the spectrum for Route 1. This suggests that, although there may be minimal covalent interactions in Route 2, their quantity is negligible, as compared with the extensive covalent bonding features for the in situ polymerization product (Route 1). The major interactions in Route 2 are predominantly non-covalent due to favourable donor-acceptor interactions between chitosan and polyaniline, as supported by the IR spectral data. While covalent bonding cannot be ruled out, there is several results that strongly support that noncovalent contributions due to complex formation cannot be ruled out altogether. A small paragraph was added to discuss the signal at 30.3 ppm for Route 2.

-        Overall I am convinced that some small covalent can be obtaining also with the route 2.

Response:

We have added this information to Section 3.1.2 as part of the discussion to address the reviewer comment.

-        Final observartion: Try to compress the abstract by keeping just the main information.

Response:

To address the reviewer comment, the abstract was compressed to ca. 200 words, while still maintaining the key information.

General response to Reviewer 1:
The authors sincerely appreciate your insightful and constructive comments on this manuscript. In turn, we have addressed all comments in a point-by-point fashion, as reflected in the revised manuscript. These constructive comments are invaluable towards improving the quality the revised manuscript and we appreciate the opportunity to improve this research contribution.

Reviewer 2 Report

Comments and Suggestions for Authors

In the manuscript entitled “Chitosan-Polyaniline (Bio)Polymer Hybrids by Two Pathways: A Tale of Two Biocomposites” The chitosan-polyaniline composite was prepared using three different methods, and the characteristic chemical interactions between polyaniline and chitosan were analyzed to determine the nature of the bond (covalent versus non-covalent). The paper has significant potential and contributes valuable insights to the field. Addressing the suggested improvements will enhance the quality of the manuscript. I recommend a Major revision.

1.      Three methods for preparing the Polyaniline and chitosan composite are mentioned in the manuscript. However, the abstract (line 13) only refers to two methods. Please resolve this discrepancy to ensure consistency between the abstract and the manuscript.

2.      Please use shorter and more specialized keywords relevant to the manuscript.

3.      Please define them just once when the abbreviation was used for the first time. Also, it is recommended that after introducing the abbreviation in the text, you use just the abbreviation, not the complete name. for example, “Chitosan”.

4.      The introduction lacks a thorough discussion of the issues at hand and does not provide a comprehensive background on previous research.

5.      Please incorporate more recent references in the manuscript. Many of the current citations are outdated. Also include other recent material characterization techniques, for example use and cite: doi.org/10.1016/j.molliq.2024.124639

6.      What’s novelty of the study? Please clarify

7.      In line 321, it is stated that Figures 4 (a-b) show only chitosan and polyaniline. However, Figure 4a is the only one that displays this. Please correct this discrepancy.

8.      In line 341, it is stated that Figure 4c shows the sample with PANI 25%. However, this sample is shown in Figure 4b. Additionally, lines 341-343 are also related to Figure 4b.

9.      The conclusion part does not highlight the importance of the findings of the present study and lacks the proper structure of a standard conclusion.

Author Response

Reviewer #2

In the manuscript entitled “Chitosan-Polyaniline (Bio)Polymer Hybrids by Two Pathways: A Tale of Two Biocomposites” The chitosan-polyaniline composite was prepared using three different methods, and the characteristic chemical interactions between polyaniline and chitosan were analyzed to determine the nature of the bond (covalent versus non-covalent). The paper has significant potential and contributes valuable insights to the field. Addressing the suggested improvements will enhance the quality of the manuscript. I recommend a Major revision.

  1. Three methods for preparing the Polyaniline and chitosan composite are mentioned in the manuscript. However, the abstract (line 13) only refers to two methods. Please resolve this discrepancy to ensure consistency between the abstract and the manuscript.

Response:

The original discrepancy related to general categorization of Method 1 as chemical (covalent) bonding, whereas Methods 2 &3 were viewed as molecular association (noncovalent bonding) based on the predominant physical processes for Routes 2 &3. To avoid this discrepancy, the abstract was rewritten so that now it contains the information about all three methods, which can largely be grouped as two categories of interactions (covalent vs. noncovalent) composites, in line with the title of this Manuscript and the discussion that follows for the Types 1, 2, and 3 composites.

  1. Please use shorter and more specialized keywords relevant to the manuscript.

Response:

The keywords have been adjusted accordingly.

  1. Please define them just once when the abbreviation was used for the first time. Also, it is recommended that after introducing the abbreviation in the text, you use just the abbreviation, not the complete name. for example, “Chitosan”.

Response:

To address this comment, all the terms mentioning polyaniline and chitosan have been converted to PANI and CHT after first introduction, accordingly.

  1. The introduction lacks a thorough discussion of the issues at hand and does not provide a comprehensive background on previous research.

Response:  To address this comment, two paragraphs on the role of chitosan were added to the introduction: 1) as a noncovalent template that serves to alter the morphology of PANI, and 2) as a reactant with PANI that undergoes copolymer formation with PANI, which may also affect the resulting composite structure due to greater entanglement between CHT and PANI. The corresponding references were added as well.

  1. Please incorporate more recent references in the manuscript. Many of the current citations are outdated. Also include other recent material characterization techniques, for example use and cite: doi.org/10.1016/j.molliq.2024.124639

Response:  Thank you for your suggestion to incorporate more recent references.  The reference you recommended is highly valuable, and we have included it in the manuscript along with a relevant paragraph that highlights its findings.

  1. What’s novelty of the study? Please clarify

Response:

The novelty lies in the exploration and comparison of three distinct preparative routes for PANI/CHT composites. Unlike previous research, which typically focuses on a single synthetic method, our study systematically investigates how these different routes influence the formation and type of bonding between PANI and CHT. Insight on this aspect cannot be fully understood via a single preparative route, in the absence of comparisons as reported herein according to use of complementary methods that range from thermodynamic to spectroscopic techniques. A corresponding discussion paragraph was added to the introduction section to address the reviewer comment.

  1. In line 321, it is stated that Figures 4 (a-b) show only chitosan and polyaniline. However, Figure 4a is the only one that displays this. Please correct this discrepancy.

Response:

The corresponding discrepancy outlined by the reviewer was corrected (the numbering of Figure 4a was changed to Figure 8a).

  1. In line 341, it is stated that Figure 4c shows the sample with PANI 25%. However, this sample is shown in Figure 4b. Additionally, lines 341-343 are also related to Figure 4b.

Response:

Corrected. Referred to Figure 8b (ex-4b) at the end of the line.

  1. The conclusion part does not highlight the importance of the findings of the present study and lacks the proper structure of a standard conclusion.

Response:

The conclusion was rewritten according to standard structured format.

General response to Reviewer 2:

The authors greatly appreciate your careful review of their manuscript and your valuable suggestions. We have addressed all comments in a point-by-point response, as reflected in the revised manuscript. The insightful and constructive comments by the reviewer have led to significant improvements and the overall clarity and impact of this contribution.

Round 2

Reviewer 2 Report

Comments and Suggestions for Authors

Dear authors,
Thank you for the revised version.